# Droplets in underlying chemical communication recreate cell interaction behaviors

Agustin D. Pizarro [1], Claudio L. A. Berli [2], Galo J. A. A. Soler-Illia [1] & Martín G. Bellino [3✉]

The sensory-motor interaction is a hallmark of living systems. However, developing inanimate systems with "recognize and attack" abilities remains challenging. On the other hand, controlling the inter-droplet dynamics on surfaces is key in microengineering and biomedical applications. We show here that a pair of droplets can become intelligently interactive (chemospecific stimulus-response inter-droplet autonomous operation) when placed on a nanoporous thin film surface. We find an attacker-victim-like non-reciprocal interaction between spatially separated droplets leading to an only-in-one shape instability that triggers a drop projection to selectively couple, resembling cellular phenomenologies such as pseudopod emission and phagocytic-like functions. The nanopore-driven underlying communication and associated chemical activity are the main physical ingredients behind the observed behavior. Our results reveal that basic features found in many living cell types can emerge from a simple two-droplet framework. This work is a promising step towards the design of microfluidic smart robotics and for origin-of-life protocell models.

[1] Instituto de Nanosistemas, UNSAM-CONICET, Av. 25 de Mayo 1021, 1650 San Martín, Buenos Aires, Argentina. [2] INTEC (Universidad Nacional del Litoral-CONICET) Predio CCT CONICET Santa Fe, RN 168, 3000 Santa Fe, Argentina. [3] Instituto de Nanociencia y Nanotecnología (CNEA-CONICET), Av. Gral. Paz 1499, San Martín, Buenos Aires, Argentina. ✉email: mbellino@cnea.gov.ar

Complex behaviors that involve intelligence to probe, recognize and interact are essential attributes in many fundamental processes in life, without which the reproduction and immune system actions cannot be explained. Examples are the phenomena of free-swimming sperm to a stationary egg[1] and the directed projection of pseudopods in a neutrophil phagocytic event[2]. Specific stimulus-response interplay and the ability to transform chemical energy into local mechanical work (active matter systems)[3,4] are also key properties of biointeractions[5]. Another common feature of the two-body interactions in nature is also the role differentiation such as the case of attacker and victim[6–8]. However, achieving the generation of similar sensory-motor skills that lead to such non-reciprocal interaction scenarios remains a challenge of paramount importance in man-made systems. In a separate but related thread, droplets (restricted fluid volumes) are ubiquitous in nature and technology. The design of droplet frameworks as simple machines to achieve complex tasks is an exciting wide open field[9–11]. In that context, out-of-equilibrium droplet scenarios hold great promise as systems to perform hydrodynamic transduction of chemical signals[11,12]. The interaction between droplets is gaining increasing attention as models to approach dynamic behaviors of living cells[13–15]. Active droplet interplay can even serve as minimalistic models by which to gain perspective on the circumstances required for life to begin[16–18]. The ability to control the attraction and coupling of droplets on surfaces is also important for a variety of applications, ranging from thermal management technologies to microfluidic liquid handling[19–22]. Droplet microfluidics on planar substrates in fact has experienced a meteoric development[23,24]. Existing strategies for the management of droplets on surfaces beyond those triggered by external stimuli (electric, magnetic, luminic)[21,22,25–28] are based either in prepared surfaces having a physico-chemical anisotropy[19,29–34] or vapour emanated by nearby droplets of carefully balanced binary composition[20]. Active droplets that interact and act by themselves via messages through a surface are eagerly awaited. The finding of a support surface able to mediate reconfigurable inter-droplet communication to autonomously execute complex dynamic tasks would indeed be a meaningful step in the evolution of practical tools for many top emerging interdisciplinary fields. Thin nanoporous layers offer unique opportunities for many practical applications owing to their fascinating properties and high tunability[35,36]. Features of imbibition at the nanoporous thin film level, such as balance between capillary infiltration and liquid evaporation[37–39], lead to peculiar transport phenomena of fluids across the nanopore network, thereby generating a steady-state annulus of infiltrated porous material around droplets lying on a nanoporous thin film surface. We consequently reasoned that such droplets may act as emitters of surface chemical messages (i.e., endowed with a peripheral signaling annulus) and thus be remotely noticed by a counter-droplet. In turn, a neighboring chemically complementary droplet could evolve transductional chemo-mechanical complex behaviors (becoming locally active) in response to the nanopore-driven annular cues from the emitting droplet. This concept of an underlying nanoporous layer to mediate an intelligent droplet interplay (chemospecific inter-droplet role-differenced autonomous operation - sketched in Fig. 1) indeed successfully works as described in the present paper.

Here we show chemo-active attacker-victim-like non-reciprocal interactions between spatially separated droplets enabled by a self-generated stimulus-response behavior when both are placed on a nanoporous thin film surface. We find that an active droplet can sense and subsequently respond with a macroscopic morphological shift and thereby coupling with a specific neighboring droplet, making use of the fluid released from the counter-droplet across the nanoporous surface as both signaling and power source. The specific droplets interplay exploits an attraction that stems from the feedback between nanopore-mediated communication and chemical activity (catalyzed decomposition of $H_2O_2$) as driving force for autonomous operation. The droplet platform that we propose and demonstrate in this study can indeed perform crucial chained actions intrinsic to living systems (sense-respond-attack), recreating even typical cell behaviors such as pseudopod projection and phagocytic actions.

## Results

**Emergence of droplet interplay on a nanoporous thin film surface.** Droplets lying onto a nanoporous thin film surface present a typical hemispherical sessile drop shape, and also form an annulus of wetted pores around the droplets[37], as shown in Fig. 2 and Supplementary Fig. 1 for a 1 wt% KI aqueous solution. The surface employed consists of a nanoporous titania thin film (180 nm thick – 45% pore volume) produced via a sol−gel method combined with template self-assembly on silicon substrates[40], presenting pore and neck sizes of ≈ 12 and 4.5 nm, respectively (see Supplementary Fig. 2 for film characterization data).

The motion of liquids in nanopore matrices is propelled by capillarity in which the advancing flow usually follows the classic Lucas−Washburn law, which predicts a square-root of time imbibition kinetics[41]. Additionally nanoporous thin film imbibition is strongly influenced by evaporation, and hence, the kinematics is different to that generally found on conventional nanoporous structures[37]. In the case of a sessile droplet on a nanoporous thin film, since capillary filling from the drop and surface evaporation reach a balance, the capillary infiltration is arrested at a given distance and a steady annular region of the wetted material is formed in the drop periphery (see Fig. 2). The fluid-filled region that is generated into the nanopore network around the droplets can be clearly distinguished thanks to the changes in the thin film interference of reflected light, because it produces a refractive index contrast in relation to the outer empty pores region. The kinematics of the infiltration−evaporation process on these nanostructures, represented by the time-dependent width of the annular wetted region, is predicted by $w(t) = (c_I \tau)^{1/2} [1 - \exp(-2t/\tau)]^{1/2}$, which accounts for the balance between capillary infiltration from the droplet (characterized by the coefficient $c_I$) and the rate of liquid evaporation to the ambient (characterized by the time $\tau$)[37]. The prediction of this equation is plotted in Fig. 2. In order to better rationalize the arrested infiltration into nanoporous thin film structures, it is worth noticing that the steady-state width of the wetted ring around the drop ($w_{ss}$) thus obeys $w_{ss} = (c_I \tau)^{1/2}$. The arrest of the infiltration is not usually found in the conventional porous structures, where evaporation has negligible effects and capillary filling expands indefinitely through the sample[42].

When a droplet of 30 wt% hydrogen peroxide solution is placed near the KI drop, we evidence a behavior which can be figuratively envisaged as a droplet attacking after the perception of the presence of the victim drop. Both droplets contain separately the chemical species for the well-known peroxide decomposition, which is catalyzed by iodide[43]. As revealed by the snapshots presented in Fig. 3 and Supplementary Movie 1, the $H_2O_2$ droplet initially spreads with the typical circular shape until it reaches equilibrium. A distinctive behavior occurs when the drop "notices" the KI droplet in the vicinity and then extends a protrusion as a leading edge (qualitatively resembling the 'pseudopods' observed in cell morphologies[44]). The $H_2O_2$ droplet indeed takes action in response to the fluid released to the

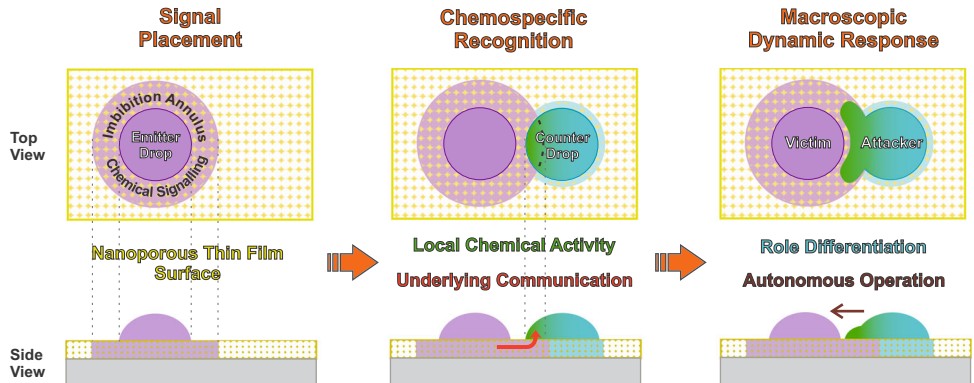

**Fig. 1 Schematic concept of the underlying nanoporous layer mediation to achieve complex inter-droplet responses.** When chemically complementary drops are deposited together onto a nanoporous thin film, they interact and subsequently act by themselves via underlying chemical messages and localized activity, leading to a macroscopic response. These droplets can spontaneously evolve to a chemospecific stimulus-response operation, which resembles an emergent "intelligent" behavior. The triggered autonomous attacker-victim-like non-reciprocal interactions lead to the generation of distinctive droplet dynamics with shape-transformation and complex behaviors. Note that the schemes are out of scale (e.g., the thickness of the nanoporous thin film is about a few hundred nanometers, which is much smaller than the droplet height).

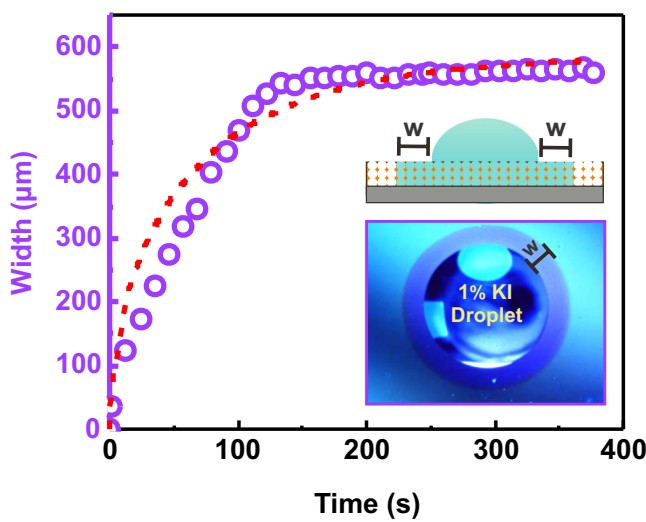

**Fig. 2 Typical nanofluidic infiltration feature when a droplet is placed on a nanoporous thin film.** Width (w) of the self-infiltrated annulus region (see sketch illustration) around a 1 wt% KI aqueous droplet deposited on the nanoporous titania thin film as a function of time (T = 25 °C and 45% RH). Symbols represent the experimental data and the red dotted line is the prediction from the kinematic infiltration—evaporation equation with $c_I =$ 1800 µm²s⁻¹ and $\tau = 175$ s. (see text for details). Note the capillary infiltration arrested at a given annulus width, resulting from the competition between the infiltration and the evaporation processes. The inset shows an optical picture of the annulus in the periphery of the sessile KI droplet.

nanopores by neighboring KI droplet that self-generate a surrounding attractor region. The protrusion then grows from the $H_2O_2$ drop and its size increases gradually up to meet the sessile KI droplet, eventually bridging the droplets. It is noteworthy that in the absence of the $H_2O_2$ drop response, the coupling between the spatially separated droplets would not have occurred (note the perimeter extrapolation corresponding to the typical circular drop shape in Fig. 3a fifth panel). As expected, no interaction was observed when the KI and $H_2O_2$ droplets were placed onto the bare silicon substrate (in the absence of the nanoporous film coating) or onto a dense titania thin film (in the absence of template) in equivalent conditions to Fig. 3 (see Supplementary Fig. 3 and Supplementary Fig. 4 for silicon

substrate only and dense titania film, respectively). These results evidence that the nanoporous layer is key to droplet interplay. Analogous experiments to Fig. 3 performed by replacing the $H_2O_2$ droplet for a pure water droplet showed no attraction between both droplets, where the peripheral annulus even behaved as a barrier for the water drop spreading (see Supplementary Movie 2). A similar "barrier" effect on the neighboring placed $H_2O_2$ droplet was also observed when the KI droplet was replaced with an equivalent 1 wt% KCl solution (see Supplementary Movie 3). These control experiments indeed illustrate a transition from an attractor to a shield behavior of the signaling annulus by changing the chemical droplet composition that clearly reveals the chemospecific stimulus-response remote interplay between the droplets on the nanoporous surface.

In order to present the spatial motion of the advancing protrusion in a more quantitative way, we tracked a material point at the protrusion front and plotted its position in Fig. 3b. The data clearly illustrates that the tracking point initially has a forward motion and then asymptotically reaches a constant value until the droplet coupling event. We also show in Fig. 3b the speed of protrusion advance as a function of time. Finally, in Fig. 3c, the temporal dependence of protrusion expanding area is plotted against the time to further illustrate the kinetics of growth. It can be seen that, as the protrusion grows in size, a decrease in the growth rate takes place. However, the protrusion tends to smoothly extend in the lateral directions as the leading edge slows down. Taken together, the dynamics of the inter-droplet connection can be divided into 'protrusion nucleation-growth' and 'coupling standby' stages. The nanoporous surface was capable of enabling an autonomous interaction between chemically complementary droplets.

**Origin of the droplet response**. The fact that droplets tend to become unstable and deform is an unusual behavior, because contact line pinning opposes such shape changes. An instability of the droplet shape indeed requires non-equilibrium conditions[16]. On the other hand, a droplet can become chemically active if species into the droplet are produced and destroyed by chemical reactions. In such a scenario, we propose that these non-equilibrium conditions in our system are provided by the chemical energy input generated by catalytic peroxide decomposition. In other words, the origin of the droplet projection action can be traced back to a local chemical activity, a simplified view of which is sketched in Fig. 4. When and where the $H_2O_2$

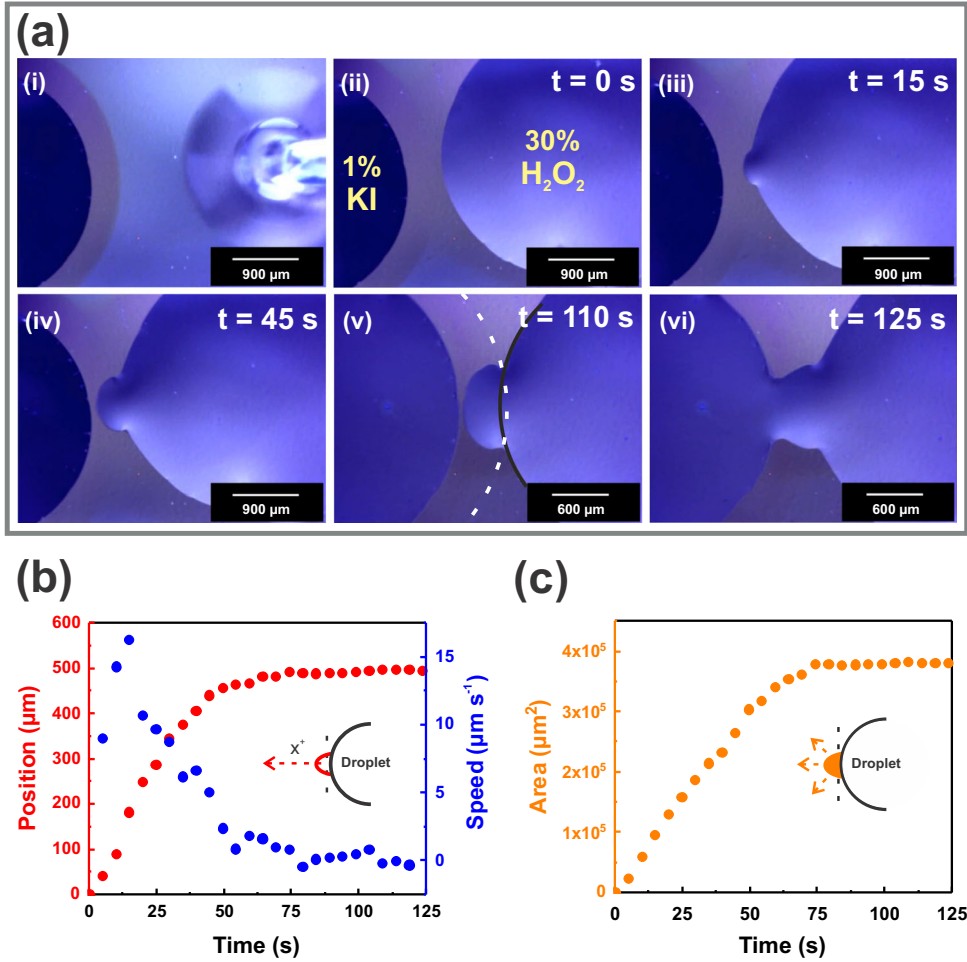

**Fig. 3 Droplet bridging dynamics. a** Snapshots to show views of the $H_2O_2$ droplet autonomously projecting by interacting with the spatially separated KI droplet on a nanoporous thin film surface (see also Supplementary Movie 1). After reaching maximum spreading and forming a circular shape, the $H_2O_2$ droplet spontaneously deforms where it notices the fluid released from the neighboring KI droplet across the nanopores (attractor annulus), followed by the projection in a protrusion morphology resembling a cell pseudopod. The protrusion then grows directionally towards the KI droplet; finally, coupling of the droplets occurs, forming a communicating bridge that allows the fast passage of material from one droplet to each other. The black and white superimposed curves in the fifth panel (v) mark the extrapolation of the outlines of the $H_2O_2$ droplet and the annulus surrounding the KI droplet, respectively. **b** Position and velocity of the tracking point of the protrusion at the leading edge as a function of time. **c** Temporal dependence of the expanding protrusion area. Protrusion decreased growth speed as it approached the KI droplet. Time interval between the dots in the curves is 5 s.

droplet comes into contact with the nanopore-carried KI-releasing from the counter-droplet, the well-known catalytic conversion of hydrogen peroxide to water and molecular oxygen occurs: $2H_2O_2(l) \rightarrow H_2O(l) + O_2(g)$. The reaction is spontaneous, exothermic and catalyzed by iodide[43]. The active species of the reactor droplet ($H_2O_2$) therefore degrades locally and abruptly into lower energy components that are soluble in the background fluid, because of the ions incorporated from the nearby catalyst droplet (KI) through the underlying nanopore network. The local chemical activity then supplies the energy to focusedly trigger a shape instability that leads to the generation and subsequent growth of the protrusion, converting chemical to mechanical energy.

To roughly quantify the energy exchanged, we compared the chemical energy released by KI-catalyzed $H_2O_2$ decomposition with the energy expenditure required for protrusion occurrence estimated from the work of adhesion of the interface. At equilibrium, the force balance of interfacial tensions at the three-phases contact line is described by the Young equation, $\gamma_{sg} = \gamma_{ls} + \gamma_{lg}\cos\theta_c$, where the subscripts stand for solid-gas, liquid-solid, and liquid-gas, respectively, and $\theta_c$ is the contact

angle[45]. In this theoretical context, one may further include the Dupre equation, which defines the work of adhesion (per unit area) of the interface formed between the solid substrate and the liquid drop: $W = \gamma_{sg} + \gamma_{lg} - \gamma_{ls}$[46]. By combining these equations, one obtains the adhesion energy in terms of the variables that can be readily measured; that is, $W = \gamma_{lg}(1 + \cos\theta_c)$. Regarding the new interfacial area progressively generated due to protrusion growth, $A_p$ (which is quantified in Fig. 3c), a simple calculation indicates that the energy change associated with this wetting event is $\Delta E_w = W A_p$. On the other hand, the energy released by chemical activity can be estimated considering the protrusion as an expanding microreactor containing the fuel ($H_2O_2$), where the decomposition reaction can gradually progress in the course of protrusion growth. We can therefore estimate the chemical energy capable of being released during protrusion evolution ($E_{ch}$), as a first approximation, using the Gibbs free energy change of $H_2O_2$ decomposition ($\Delta G$), $H_2O_2$ concentration ($C$), and a reaction volume proportional to $A_p$ according to the $V_d/A_d$ ratio, where $V_d$ and $A_d$ are the volume and contact area of the drop respectively; which results in $E_{ch} = \Delta G C A_p V_d/A_d$. The comparative value of $E_{ch}/\Delta E_w \approx 10^7$ that is characteristic throughout

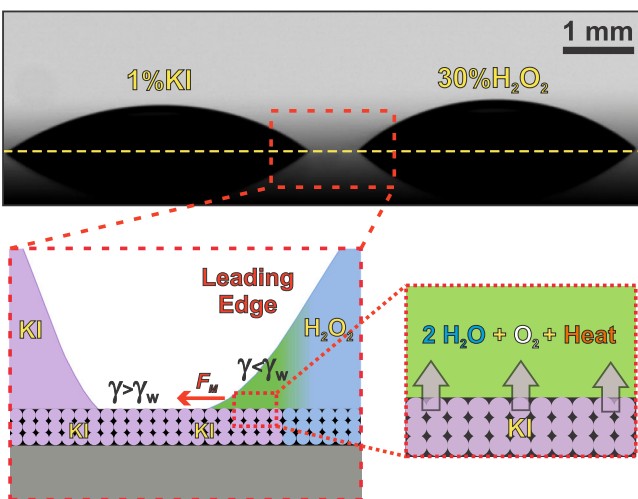

**Fig. 4 Schematic representation of the underlying nanopore gating for macroscopic liquid movement.** The picture shows a side-view of the interacting KI and $H_2O_2$ droplets after placed on the nanoporous surface. The zoom schematics illustrate the locally active droplet resulting from the nanopore-mediated underlying communication with the neighboring droplet. The catalyzed decomposition of hydrogen peroxide takes place in the region where a fraction of the $H_2O_2$ droplet perimeter overlaps the annulus of the KI droplet. The droplet can locally escape from the steady-state shape if the energy associated with chemical reaction overcomes the pinning energy barrier. The temperature increases in the reaction zone, $O_2$ concentration follows the same trend and in turn the $H_2O_2$ concentration decreases. This chemical and thermal scenario leads to localized variations of the surface tension, which induce the Marangoni stress $F_M$ (tangential force per unit area) that drag the liquid from the $H_2O_2$ droplet towards the KI droplet (red arrow), in the direction parallel to the substrate, over the KI annulus.

protrusion growth - for $\gamma_{lg} \approx 0.074$ N/m[47], $\theta_c \approx 35°$ (see picture in Fig. 4), $\Delta G = -116.8$ kJ/mol (calculated from the individual standard Gibbs free energies of formation), 30 wt% $H_2O_2$ solution and $V_d/A_d \approx 10^{-4}$ m (considering $V_d = 2$ µl and a $A_d$ estimated by extrapolation from Fig. 3) - clearly indicates that in principle there is extremely more than enough chemical energy available to fuel the protrusion developing.

Since the catalyst (iodide) can perform many rounds of catalysis and the reaction rate is relatively high for the catalyzed decomposition of hydrogen peroxide, the limiting factor for power generation rate can be inferred a priori from species diffusion in the reaction zone. However, considering the short lengths involved (say $\Delta z = 100$ µm, as typical scale) and the large diffusivity of the chemical species (for example the diffusion coefficient of $H_2O_2$ is $D \approx 1.5 \ 10^{-9}$ m²s⁻¹[48], the characteristic diffusion times results $\frac{(\Delta z)^2}{2D} \approx 3$ s, which evidences that the reaction can gradually proceed during the protrusion evolution. It is known that molecular diffusion across nanopore networks can decrease up to an order of magnitude with respect to the bulk value due to confinement effects, owing to the relevance of interfacial molecule-surface interactions[49,50]. In this limiting case, the estimated timescale of the diffusion can potentially increase to 30 s. This scenario still supports reaction progress during protrusion development and incidentally the slow protrusion kinetics is in line with the fact that confinement-induced effects on molecular diffusion might be present. Also related to the large amount of energy that might be released by the reaction, which would substantially increase the liquid temperature, one should note that the relatively large thermal diffusivity of water ($\alpha \approx 1.43 \ 10^{-7}$ m²s⁻¹)[51] yields a small value of the characteristic time for

thermal diffusion, $\frac{(\Delta z)^2}{2\alpha} \approx 0.03$ s. This result, which is typical of microscale problems, means that the generated heat can be quickly dissipated while the reaction proceeds gradually, so while local heating is contemplated, large temperature jumps are not expected. The simple energetic approach proposed here predicts feasible the protrusion deployment powered by the catalytic reaction arising from the nanopore-mediated communication between the droplets. This locally active droplet can indeed overcome the pinning energy barrier to escape from the original shape and develop the protrusion until it meets the sessile KI droplet. Protrusion next undergoes a standby stage until the inter-droplet interfacial tension is broken to achieve the bridging junction.

In the previous paragraphs we have assessed the elemental thermodynamic variables to prove that the free energy locally released by the chemical reaction provides enough energy to feed the wetting process involved in the protrusion formation. In what follows we employ physicochemical hydrodynamics to interpret the localized motion of the triple contact line. The main hypothesis is that the driving force that promotes the advance of the liquid front is the Marangoni effect[52], which originates at the $H_2O_2$-drop contact line, where the liquid surface tension $\gamma$ is reduced by the chemical reaction. The generated surface tension gradient ($d\gamma/dx$, being $x$ the direction from the $H_2O_2$-drop towards the KI-drop) yields a tangential force per unit area on the air-liquid interface (the Marangoni stress; see the schematic representation in Fig. 4)[53], which pulls the $H_2O_2$-droplet contact line outwards and originates the protrusion. The progress of the reaction preserves the surface tension difference and the protrusion keeps growing.

Droplet spreading by Marangoni effect has been well described in the literature[54,55]. Here we inspect how the chemical reaction generates the gradient $d\gamma/dx$ in our system. The variation of $\gamma$ has two contributions: $d\gamma = (\partial\gamma/\partial C)dC + (\partial\gamma/\partial T)dT$, where the first term comes from the local change of substances concentration ($C$) and the second one comes from local temperature ($T$) variation. Concerning the latter, it is worth to recall that $\gamma$ always decreases with increasing liquid temperature[53]. For example, for pure water, $\gamma_w \approx 73$ mN m⁻¹ at 20 °C and $\gamma_w \approx 71$ mN m⁻¹ at 30 °C. Therefore, as the reaction of $H_2O_2$ decomposition is exothermic, the generated heat in the reaction zone induces a surface tension difference with the surrounding fluids. Regarding the solutal contribution, we take into account the main components of the reactive system: (i) The presence of $H_2O_2$ slightly increases the surface tension of the solution in relation to pure water (73.7 mNm⁻¹ for $H_2O_2$ 28% at 18 °C)[56], however the compound is consumed by the reaction. (ii) The presence of dissolved oxygen decreases the surface tension of water (68 mNm⁻¹ for 45 ppm at 20 °C)[57], and the effect is stronger under supersaturated conditions[58]. In contrast, the surface tension of KI solution is slightly larger than that of water ($\gamma_K \approx 74$ mN m⁻¹ for KI 1 M at 20 °C)[59], and it may further increase in the annulus due to evaporation-induced concentration. Lastly, it is relevant to point out that both thermal and solutal contributions have the same direction: $d\gamma/dx > 0$ from the reaction zone towards the KI-droplet.

Summarizing, when the chemical reaction starts, $\gamma$ locally decreases and generates a Marangoni stress towards the regions where $\gamma$ is higher (KI droplet). The Marangoni stress drags fluid from the $H_2O_2$-drop and drives the contact line motion (see Fig. 4). Solving the full hydrodynamical problem requires calculating the time-dependent concentration and temperature fields in the reaction zone, coupled to fluid momentum equations, which demands numerical simulations. Alternatively, here we apply scaling analysis to extract basic relations that account for the fluid front motion. We propose that the Marangoni flow pulls

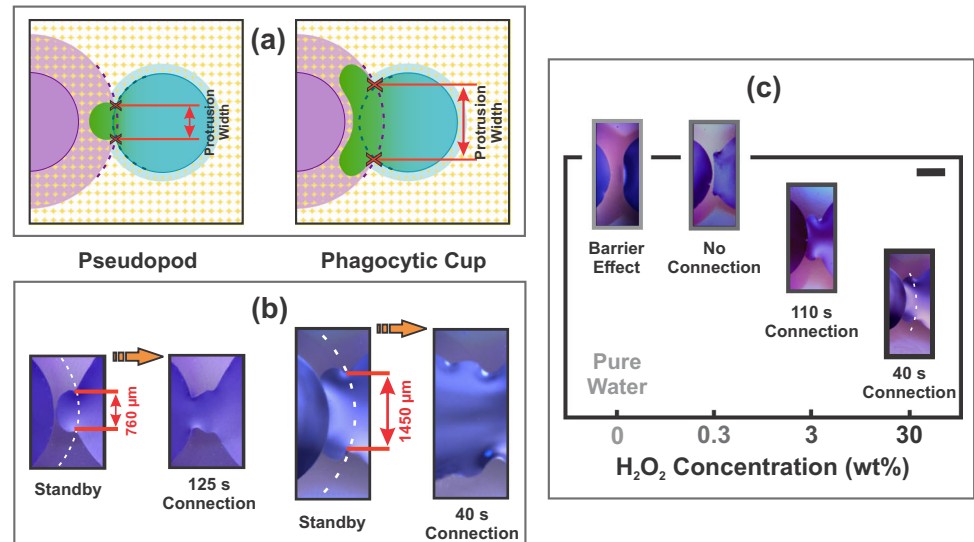

**Fig. 5 Programmable interplay of droplets. a** Geometric illustration of different protrusion widths achieved as a result of the different overlaps reached by the $H_2O_2$ droplets with the annulus of KI droplets. **b** Pictures just before and after bridging between 30 wt% $H_2O_2$ and 1 wt% KI droplets to illustrate the distinct protrusion widths and morphologies. The dashed white superimposed curves mark the extrapolation of the annulus surrounding the KI droplet. Note that a more widened protrusion shape is indeed observed due to a greater crossing between the attacking $H_2O_2$ droplet edge (right) and the annular signaling from the KI droplet (left). The time just after droplet coupling indicate the different connection times between the droplets that are linked to the different protrusion widths. **c** Dependence of the interaction behavior between $H_2O_2$ droplets of different concentrations and 1 wt% KI droplets. Insets illustrate the standby stage for the different $H_2O_2$ concentrations. Scale bar = 1 mm. A longer time from protrusion nucleation to droplet coupling (connection time) was observed in the case of 3 wt% $H_2O_2$ droplet with respect to the 30 wt% $H_2O_2$ droplet concentration. The 0.3 wt% $H_2O_2$ droplet showed the protrusion generation step but failed to couple. Instead, a barrier effect was found in the case of the water droplet, where the signaling annulus coming from the KI droplet even repelled the total spreading of the water droplet.

the droplet contact line with a velocity on the order of $u_x$, so that the position $x_f$ of the advancing front moves as $dx_f/dt = u_x$. As a first approximation, we consider the primary edge of the advancing fluid front as a thin fluid layer of thickness $h$ that is subjected to Marangoni stress in the $x$-direction and then the fluid velocity at the liquid surface results in $u_x = (d\gamma/dx)h/\mu$, where $\mu$ is the fluid viscosity[53,60]. One may further assume a nearly constant surface tension difference $\Delta\gamma$ along the flow path, $d\gamma/dx \approx \Delta\gamma/x_f$[55,60], and thus the motion equation is written $x_f dx_f/dt = \Delta\gamma/\mu$. After integration with the initial condition $x_f = 0, t = t_0$, the front position results in $x_f(t) = \left[c_M(t - t_0)\right]^{1/2}$, where $c_M = 2h\Delta\gamma/\mu$ is here defined as the Marangoni flow coefficient. Also with this nomenclature, the fluid front velocity is $dx_f/dt = c_M/x_f(t) = \left[c_M/4(t - t_0)\right]^{1/2}$.

The model prediction is checked against experimental data in Supplementary Fig. 5a (position) and Supplementary Fig. 5b (velocity). It is readily seen that the square-root-of-time kinematics closely follows the time variation of the protrusion front position, with $c_M = 5396\ \mu m^2\ s^{-1}$ and $t_0 = 8.5\ s$. It is relevant to note that we have arbitrarily chosen $x_{f0} = 0$ at the start of the position tracking. Then the existence of a sort of initial induction $(t_0)$ is associated to the time required to reach the Marangoni flow regime (formation stage of the concentration and temperature gradients). This induction time is more clearly observed in Supplementary Fig. 5b, where the experimental velocity initially raises (from still to being in motion), as physically expected in the transitional stage, and then decreases as predicted by the model (note that the driving force $\Delta\gamma/x_f$ of the contact line displacement decreases with the advance of the front position). Therefore, the model applies in the time range $10 \leq t \leq 50 s$ characteristic of the protrusion growth stage, approximately. The upper limit is indeed related to the time at which the protrusion meets the KI-droplet and the front position

reaches a long-lasting pseudo stationary state, before drop coupling. Regarding the magnitude of the coefficient $c_M = 2h\Delta\gamma/\mu$, if one considers, for instance, $\Delta\gamma \approx 3\ mN\ m^{-1}$ and the viscosity of water at room temperature (1 mPas), one founds that $h \approx 1\ nm$ (thickness of the edge in the advancing contact line front) is enough to explain the macroscopic effect. Since these parameter values are quite reasonable for the system[61], one may conclude that both the functionality and the quantitative scale of the model satisfactorily agree with the observed wetting phenomena.

**Versatility of inter-droplet operation**. We next illustrate that droplet interaction can be programmed by controlling simple system variables. The protrusion size is related to the overlapping between the attacking droplet edge and the annular signaling from the counter droplet (see together Fig. 5a, b). The protrusion base at the standby stage encompasses on the rim of the $H_2O_2$ droplet among the two points where it intersects the KI annulus. The distance between these two points of intersection determines the hereafter named protrusion width. The different overlaps reached by the $H_2O_2$ droplets with the annulus of the KI droplets thus produce different width of the protrusions (see Fig. 5a, b), which in fact is geometrically determined by the distance between the centers of the droplet circumferences[62]. One may even find a protruded shape that resembles a 'phagocytic cup' morphology[63] when the overlapping is larger (see Fig. 5b). Likewise, the connection time between the droplets (i.e., the time from protrusion nucleation to droplet coupling) is linked to the protrusion width: increasing two-fold the width of the protrusion reduces droplet connection time from 125 to 40 s (see Fig. 5b).

Interestingly, Fig. 5c also shows that the inter-droplet connection strongly depends on the $H_2O_2$ concentration: droplets of 3 wt% $H_2O_2$ concentration have an about 3 times increase in the droplets connection time with respect to the 30 wt% $H_2O_2$

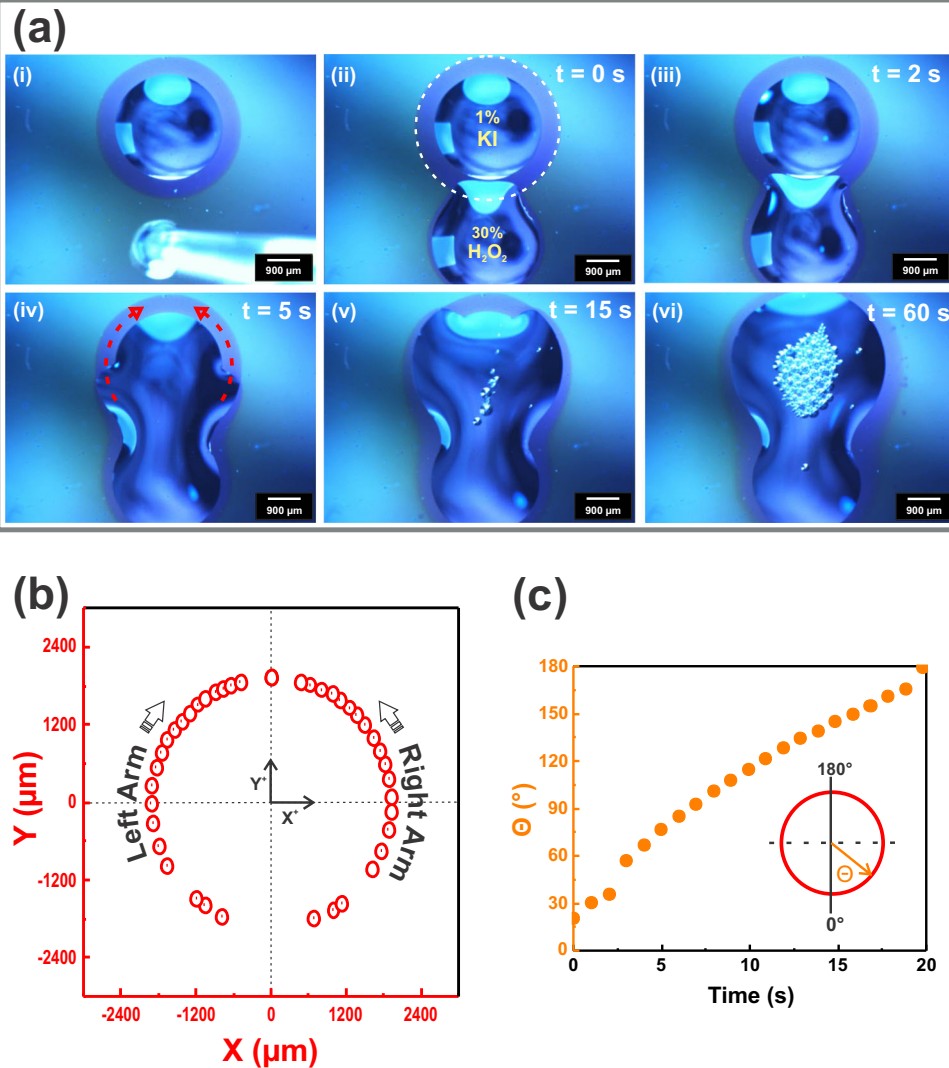

**Fig. 6 Droplet engulfing dynamics. a** Sequenced images to show a droplet engulf droplet behavior (see also Supplementary Movie 4). The $H_2O_2$ droplet extends a "phagocytic cup" in response to interaction with the fluid released through the nanopores from the KI droplet. A growth of projected arms is then directed to surround the KI droplet and finally the tips of the arms meet and fuse to engulf it. The dotted white line in the second panel marks the extrapolation of the contours of the guiding annulus self-generated in the KI droplet periphery. The red arrows in the fourth panel illustrate the path followed by the outstretched arms of the $H_2O_2$ droplet maneuvering around the KI droplet. **b** Two-dimensional trajectory of the arms extension. **c** Dependence of the rotational orientation angle of the right arm (θ) against the time. The arms are driven to rotate in a directed pathway surrounding the KI droplet. Time interval between the dots in the curves is 1 s.

droplets. The 0.3 wt% $H_2O_2$ droplet also showed the protrusion generation step but without achieving the coupling outcome. The decreased affinity between droplets with decreasing $H_2O_2$ points in the same direction as before: the chemical activity that induces active stress seems to be the main driving force towards droplet interaction.

Our minimal platform also supports the formation of other dynamical regimes of attacker-victim-like non-reciprocal inter-action typical in living cells. On widening protrusion, we observe progression to an $H_2O_2$ droplet that acts to engulf the other one (see Fig. 6a and Supplementary Movie 4). The $H_2O_2$ droplet in this case extends arms guided by the signaling annulus in the KI droplet periphery. The arms finally close at the distal region internalizing the KI droplet. Fig. 6b, c quantitatively show that the arms follow a circular pathway around the nearby KI droplet, which requires constant reorientation of the forward axis direction. This take-shape droplet dynamic is reminiscent of the phagocytic events which happen when a cell needs to engulf for

the remotion of pathogens as a core function of the immune response or for nutrition in unicellular organisms[64].

Moreover, the square-root-of-time kinematics predicted by the model proposed for protrusion advance holds all along the circular flow paths (see Supplementary Fig. 5c). Nevertheless, in this case the flow coefficient is $c_M = 1.9 \times 10^6$ $\mu m^2$ $s^{-1}$, which is 350 times larger than the one found in Supplementary Fig. 5a. This outcome means that, other factors being equal, the "effective" $h$ is larger than 300 nm for this experiment. The value is better understood if one takes into account that the inner borders of the protruding arms are merged to the KI-droplet. Hence, the relatively large values of $dx_f/dt$ can be due to extra contributions of the Marangoni effect, now acting directly from the fluid arms ($\gamma < \gamma_w$) to the sessile drop ($\gamma > \gamma_w$), across the coalescing zone of the drop perimeter. Effectively, these Marangoni-driven flows have been used to explain the fluid transfer from one droplet to another[65] and lately investigated to enhance mixing in paired droplets[66,67].

The link between the chemical activity and active $H_2O_2$ droplet motion was also clear when changing the catalyst (KI) concentration. By decreasing the KI concentration to 0.5 wt%, only a short roam of the protrusion occurred on the outline of the KI drop before coupling (see Supplementary Movie 5) and in the case of a 0.1 wt% KI droplet the "barrier" effect (locally stopped $H_2O_2$ drop spreading) was observed (see Supplementary Movie 6). The diminished action between droplets with decreasing content of the catalyst again illustrates that droplet interplay can be tuned by controlling the chemical activity. Decreasing KI concentration also caused narrower annuli at the droplets periphery. This reduction of the wetted annulus is understood from the simple model for the steady-state width $w_{ss} = (c_I \tau)^{1/2}$, considering the concentration dependence of model parameters $c_I$ and $\tau$[38]. A decrease of the salt concentration strengthens the liquid evaporation rate ($\tau$ decreases) leading to narrower wetted rings. An additional cause behind the wetted ring narrowing is attributed to a decrease in the porous matrix permeability by decreasing the ionic strength, which leads to smaller values of the infiltration coefficient $c_I$. The new infiltration-evaporation balance indeed drives to narrower annulus.

We also conducted some experiments to preliminarily explore the role of the nanopore surface modification on droplet interplay. We performed two different surface modifications through grafting of either octyltriethoxysilane (OTES) or (3-aminopropyl)triethoxysilane (APTES) (see Supplementary Fig. 6 for details). Nanopore surface modification was implemented by OTES in order to make them more hydrophobic[68] (represented by a significant increase in the film contact angle from 37° to 77°). This leads to a practically negligible annulus around the KI droplet (as expected, nanopore hydrophobicity strongly decreases the $c_I$ coefficient and therefore the imbibition was markedly diminished). Experiments carried out on OTES-modified films at inter-droplets (KI-$H_2O_2$) distances typical of macroscopic actuation occurrence on the non-modified films showed no response in both droplets (see Supplementary Fig. 7). The lack of response is indeed attributable to the practical absence of the signaling annulus around the KI droplet. In the case of APTES modification (a frequently used amino functionalization)[69], a dissimilar scenario was observed. A relatively wide annular wetted region was self-generated around the KI droplet on the APTES-modified films (only a slight change in the contact angle was observed from 37° to 46°). When a $H_2O_2$ droplet was placed near the KI drop, although a fluidic connection was attained ($H_2O_2$ drop fraction overlapping the KI annulus), solely a slow and slight elongation of the $H_2O_2$ droplet (no protrusion generation) was observed and without achieving the coupling between the droplets (see Supplementary Movie 7). This $H_2O_2$ droplet behavior is in stark contrast to the highly dynamic droplet responses arising from the non-functionalized films. Since the system can be seen as that $H_2O_2$-containing droplet chemically transducing ionic inputs driven by nanopores into mechanical actions, such change in behavior is therefore attributable to the amine-induced physicochemical transformation of the titania surface which hinders transport of ions to the $H_2O_2$ droplet across the nanopore matrix. Indeed, ion transport though nanoporous structures has been extensively studied and is well-known to be governed by the nanopore wall surface modification[40,70]. These simple surface modifications of the nanopores reveal that both fluidic and chemical communication between the droplets can be significantly affected by nanopore surface tuning and consequently alter the mechanical response of the droplets. The wide palette available of organic or metallo-organic functions permits one to anticipate that droplet emerging dynamics could be controlled at will by the custom design of nanopore surface modification. Finally, we as well show that the complex emerging dynamic between the droplets is not limited to one specific nanoporous morphology. We similarly placed KI and $H_2O_2$ droplets on a nanoporous titania thin film with smaller diameter (6 nm) pores (see Supplementary Fig. 8 and Supplementary Table 1 for film characterization data) and the $H_2O_2$-attacking behavior was observed (see Supplementary Movie 8). This straightforward experiment indeed shows that the actively interacting droplets can be also obtained in a different nanopore configuration. Taken together, the results revealed in this work illustrate a versatile, robust and flexible experimental platform of intelligently interactive droplets.

**Discussion**

We have introduced a minimal framework where droplets can autonomously experience chemospecific recognition and subsequent non-reciprocal interaction, without the need of surfaces that have guiding tracks or external stimuli. Only three actors are necessary: a water droplet with $H_2O_2$, a KI aqueous droplet and a nanoporous thin film surface. The nanoporous layer is able to mediate droplet interaction through nanopore-driven simple chemical messengers, which lead to emerging complex droplet dynamics. We showed that the $H_2O_2$ droplet can sense and selectively attack the neighboring KI droplet due to specific chemical interactions that trigger a physical response; the chemical energy is transformed into mechanical work. Figuratively speaking, the chemical signal emitted from the KI droplet (underlying nanoflow), is perceived by the $H_2O_2$ droplet and also employed to direct their attack. These interacting units indeed extract useful work from its chemical activity for an autonomous operation. The platform further supports the generation of different dynamic couplings between the droplets. The mechanism behind the droplet behaviors can be traced back to a local chemical activity that induces Marangoni stresses, which give rise to a directed droplet shape transformation. Many cell functions, such as cytoplasmic streaming, morphogenesis and motion, are driven by active stresses arising from a localized activity within the cell[71]. In particular, the cell protrusions are owing to active stress-driven actin treadmilling (polymerization) that focalizedly curve the cell membrane forward[72]. It is interesting to note that the mechanical droplet responses bear resemblance (although coming from a rudimentary way of localized active stress - chemically induced local Marangoni stress -) to protrusion emission and even phagocytic actions in biological cells. Our results indeed reveal that complex droplet dynamic behaviors reminiscent of cell morphogenesis can arise despite the relatively simple active matter framework employed. It should be also noted here that our abiotic platform brings together hallmark features of living systems, such as: specific stimulus-response interactions, role differentiation and transformation of chemical energy into mechanical work. Such active droplets could represent interacting protocells with a primitive intelligence based on simple chemical reactions in the challenging search for experimental models that explain the early steps in origin of life. We envisage that many other catalytic reactions can in principle be used as chemical complementarity in the underlying communication, which would launch the creation of a diversity of intelligently interacting droplet families. We also anticipate that the droplet behavior could be handled by nanoporous system design (i.e., framework composition, network morphology hierarchy, and pore surface functionalization). Accordingly, the droplet abilities here reported that are analogous to living systems (sense-response-interaction) lays the groundwork for the design of new microfluidic smart robotics, which will be of practical relevance for applications in chemistry, biology, physics, and beyond.

## Methods

Crack-free nanoporous titania thin films were prepared by dip-coating with a supramolecularly templated oxide precursor, following well-reported protocols[40]. The technique is based on combining sol-gel chemistry with an evaporation-induced self-assembly strategy, using an ethanol solution with an inorganic precursor (TiCl$_4$) together a polymeric template (Pluronics F127 = (EO)$_{106}$(PO)$_{70}$(EO)$_{106}$, where EO and PO represent ethylene oxide and propylene oxide blocks, respectively). Particularly, the process was performed by dip-coating silicon substrates (University wafers) in a precursor solution with TiCl$_4$:EtOH:H$_2$O:F127 = 1:40:10:0.005 molar ratios at a withdrawal rate of 3 mm s$^{-1}$ and a relative humidity (RH) of 30%. After deposition, as-prepared films were aged in a 50% RH chamber for 24 h. Next, the films were subjected to a consolidation thermal treatment that consisted in the following steps: 2 h at 60 °C, 2 h at 130 °C and finally calcined 2 h at 450 °C (to remove the templating agent) with a ramp temperature of 1 °C min$^{-1}$. Non-porous titania films were made as described but in the absence of the templating agent. In the case of smaller pore samples, the F127-loaded films were directly calcined after deposition (in the absence of the consolidation thermal steps) by a fast-firing treatment, with a dwell temperature of 450 °C and a dwell time of 10 min. In order to surface modification the nanoporous titania thin films were immersed in a 0.002 M F/toluene solution for 24 h at 60 °C (with F = OTES or APTES) followed by rinsing with toluene, EtOH sequentially, and drying with a soft stream of N$_2$ to remove the unreacted organic compounds.

In order to perform the drop interplay assays, droplet pairs of chemically different solutions of a given volume of liquid (typically 2 μL) were placed on the nanoporous surfaces from a Eppendorf automatic micropipette at T = 25 °C and 45% RH controlled ambient conditions. The growth and relative stabilization of the peripheral annulus self-generated in the first placed droplet was waited to deposit the other droplet nearby. Droplet interaction behaviors were followed with an Arcano ZTX-T trinocular stereo microscope combined with an Arcano MIC-ACC-942 10 megapixels video camera and analyzed using the Tracker software (Open Source Physics). The phenomenological droplet evolution at the different experimental conditions resulted highly reproducible.

## Data availability

The data that support the findings of this study can be found in the article and the Supplementary Information files. Any other relevant data are available from the corresponding author upon reasonable request.

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

## Acknowledgements

This work was supported by Agencia I+D+I funding (PICT 2016-1781), Argentina (MGB). Raúl Urteaga is acknowledged for image analysis. A.D.P. acknowledges his doctoral scholarship from CONICET, Argentina.

## Author contributions

M.G.B. and G.J.A.A.S-I. conceived the project. A.D.P. fabricated the samples and performed the experiments. C.L.A.B. built the model. G.J.A.A.S-I., C.L.A.B. and M.G.B. wrote the manuscript, and all authors were involved in analysis of data and discussions of this work.

## Competing interests

The authors declare no competing interests.
