## [Peer Review File · Nature Communications]

Droplets in underlying chemical communication recreate cell interaction behaviorsREVIEWER COMMENTS

Reviewer #1 (Remarks to the Author):

The authors present an experimental study on the interaction of KI and H₂O₂-containing aqueous droplets on mesoporous titania substrates. Time-dependent optical microscopy reveals remarkably complex merging dynamics and merging morphologies as a function of the droplet constituents. This complex fluidic behavior is reminiscent of the behavior of simple living systems, despite of the relative simplicity of the active matter systems studied. The experiments are carefully performed, analyzed, interpreted and related to available literature. Given the high and broad interest in experimental realizations of life-mimicking behavior and the novelty of the experimental system with respect to the nanoporous substrate wetting geometry, I think the manuscript is in principle suitable for publication in Nature Communications.

I have just two minor recommendations:

(1) The authors should clarify in a more detailed manner, why mesoporous wetting substrates are necessary for achieving the complex phenomenology. Obviously, it is caused by the peculiar imbibition front and imbibition front dynamics surrounding the droplets (annular precursor rings). However, I think the manuscript could profit from more explicit explanations with regard to the formation mechanisms of these precursor rings, beyond referring to previous work.

(2) Would it make sense to vary not only the liquid constituents of the droplets but also the pore morphology and pore size to achieve distinct droplet merging dynamics in the future?

(3) So far only the thermal diffusion time scales are estimated. It would be great to get also a quantitative or semi-quantitative understanding of the spreading and merging kinetics of the droplets. I would guess that it sensitively depends also on the fluid imbibition and fluid transport dynamics in the nanoporous films. Is it possible to relate the observed kinetics with simple models for transport of mesoporous media, e.g. based on the Darcy description?

Reviewer #2 (Remarks to the Author):

Pizarro et al report in their manuscript "Droplets in Underlying Chemical Communication Recreate Cell Interaction Behaviors" on droplet interaction mediated by an underlying mesoporous surface. This is a very timely topic of high interest to a broad readership. I would support publication in Nature Communication, however not in its present form. Normally, I don't suggest as peer reviewer additional experiments to be done. However, given the potential of this work and the high standard of Nature Communications, I would ask the authors to consider the below comments in a revised manuscript:

- The introduction is in my view too brief and does not set the scene for the state of the art and the novelty provided by this work. While the milestone papers are mentioned, more detail should be offered including a broader portfolio of relevant papers and the current knowledge gap.

- In my view, the key novelty of this paper is the fact that the underlying mesoporous layer is able to mediate droplet interaction through mesopore-driven fluid transport beyond the topological footprint of the macroscopic droplet. In my view the result shown in Figure S1 is key and should therefore be included in the main manuscript.

- While the evidence provided in Figure S1 is compelling, the extent of pore filling could be spatially mapped by either microspectroscopy (offering spectroscopic sampling of approx. 10µm sized spots under an optical microscope) or by spectroscopic ellipsometry with microspot optics (spatial resolution approx 60µm).

- In my view, it would be quite valuable to investigate the role of surface modification of the mesopores (e.g. through grafting of octadecyltrichlorosilane), which will affect the wetting of the

pores and therefore also the droplet communication. A simple control experiment would be to study droplet communication on a solid / non-porous support. Establishing the role of pore functionalization and porosity in general would also provide means for programmable surfaces.

- In my view, the authors have found a highly interesting phenomena that is worth reporting on a high profile platform. However, some of the analogies appear a little far reached. It would be quite useful to understand why the authors consider their work to be biomimetic. While I don't challenge the role of ions and reactive oxygen species in cell communication, the novelty of the work reported herein is about substrate mediation. It would be interesting if the authors could provide a better link to cellular behaviors.

Reviewer #3 (Remarks to the Author):

The work presented by the authors is an original attempt towards mimicking a basic cellular process, such as phagocytosis, via a competition between chemical reactions of two nanoconfined fluids, and the mechanical pinning energy associated with the drop stability of one of the interacting fluids when placed on a nanoporous membrane. The novelty of this study lies in simulating an attacker-victim like situation analogous to the entrapment of a protein/pathogen by a cell/neutrophil guided solely by an autonomous chemospecific interaction, without the requirement of any external stimuli or pre-existing trajectory. The authors determined conditions, in terms of concentration and the chemical nature of the constituents, yielding favorable conditions for the intended chemo-specific response. The authors have also provided a reasonable thermodynamic basis to rationalize the attacker-victim like interaction, within which the aq. H₂O₂ droplet utilizes a trajectory provided to it by the advancing annulus of the aq. KI drop.

The novelty in the scope and objectives of this study potentially makes it a fresh contribution to the existing literature in this domain, which is well cited in the manuscript. The experimental methodology used and the analysis and interpretation has a logical progression and appears clear enough for reproduction and customization. However, the investigations on the system should significantly be expanded, in terms of both experimental results and analysis. In its present state the paper lacks a clear presentation of the physical principles giving rise to the observed behavior, and would greatly be improved with e.g. a basic quantitative model explaining how to manipulate the interaction behaviors as a function of well-identified external dimensionless parameters. Most likely, a significant revision would be necessary to warrant publication in Nature Communications.

In addition to the general comments above, please find below some remarks and questions that would need to be addressed.

1. The paper would be clearer with an introductory figure showing a sketch of the system of interest with drawings of the basic processes at play (e.g. based on Figure 2) and summarizing the study. Currently, most of the figures are microscope images and/or graphs and feel rather technical.
2. The structural aspects of the porous host matrix have not been sufficiently elaborated. The existence of a "neck" and "pore" gives two indications; the "necks" being interconnections between the "pores" or the existence of ink-bottle geometries, wherein the pores get narrower ("necking") towards their openings. Could the authors provide further structural information on their porous membrane characterization?
3. The transport of liquids can be strongly modified under nanoporous confinement and the geometry of confinement also plays a significant role, for example, with introduction of anisotropy in the diffusion coefficient. Can the authors comment on whether these effects are relevant in their study (e.g. to explain the timescales involved in the experiments)?
4. What is the role played by the KI concentration? Will the phagocytic entrapment of aq. KI by aq. H₂O₂ be still observed if the aq. KI concentration is raised to 2% w/w, or reduced to 0.5% w/w for example?

5. Could the barrier resistance resulting from KCl be simply a consequence of its low concentration? Could a possible oxidation of the chloride ion to chlorine gas by H₂O₂ have led to an attacker–victim situation as well, given the concentration of KCl was an order of magnitude higher?

6. As mentioned in the manuscript, and visible from the figures and supplementary videos, the protrusion width clearly has an impact on the connection time. However, the factors leading to differences in the protrusion widths is not satisfactorily explained, and hence require further elaboration.

7. The speed of protrusion as shown in Figure 1b increases very sharply up to 13 s and then starts decreasing gradually, while no sharp changes are observed in the position. How do the authors justify it?

Please find below some additional, minor recommendations.

1. The SEM image of the porous membrane is of the 2D plane perpendicular to the pore-axis. A cross-sectional SEM/TEM image of the plane parallel to the pore axis will also be quite informative.

2. The caption for Fig 1 (a) has a lot of text that is better suited for the main text, Panels should be numbered (e.g. i, ii, iii, etc), and more precisely referred in the main text. This comment holds for all figures with multiple panels in a single figure sub-part.

3. The quantity E_{ch} needs to be defined.

4. In page 8 of the manuscript, it is mentioned that V_d/A_d is approx. equal to 10^{-4} , and the reader's attention is drawn towards Figure 1 for reference. However, looking at Figure 1 this aspect does not seem apparent. Please consider elaborating Figure 1 suitably to address this issue.

5. "Likewise, the connection time between the droplets (i.e., the time from protrusion nucleation to droplet coupling) is linked to protrusion size: for example, increasing two-fold the width of the protrusion (see Figure S3 in SI) reduces droplet connection time from 125 to 40 s (See Figure 1 and Figure 3 for the 30 % H₂O₂ sample)." (Page 10 of the manuscript). A direct reference to Figure S3 does not give any clear information about the connection time between the droplets. I suggest elaborating Figure S3 more, such that it can be directly correlated with Figures 1 and 3 in the main manuscript.

Detailed answer to the reviewers' comments:

We thank the reviewers for their constructive comments and for the time and efforts they dedicated to our manuscript, which helped us to improve our work.

In the following, the reviewers' comments are typeset in *italics* while our replies appear in regular font.

Reviewer #1 (Remarks to the Author):

The authors present an experimental study on the interaction of KI and H₂O₂-containing aqueous droplets on mesoporous titania substrates. Time-dependent optical microscopy reveals remarkably complex merging dynamics and merging morphologies as a function of the droplet constituents. This complex fluidic behavior is reminiscent of the behavior of simple living systems, despite of the relative simplicity of the active matter systems studied. The experiments are carefully performed, analyzed, interpreted and related to available literature. Given the high and broad interest in experimental realizations of life-mimicking behavior and the novelty of the experimental system with respect to the nanoporous substrate wetting geometry, I think the manuscript is in principle suitable for publication in Nature Communications.

Answer: We would like to express our acknowledgement to Reviewer #1 for his/her interest and positive opinion on the work.

I have just two minor recommendations:

(1) The authors should clarify in a more detailed manner, why mesoporous wetting substrates are necessary for achieving the complex phenomenology. Obviously, it is caused by the peculiar imbibition front and imbibition front dynamics surrounding the droplets (annular precursor rings). However, I think the manuscript could profit from more explicit explanations with regard to the formation mechanisms of these precursor rings, beyond referring to previous work.

Answer-Action taken: We thank the reviewer for this comment that helped us clarify the manuscript. As the reviewer points out, the nanoporous wetting surfaces are in fact

necessary for achieving the complex inter-droplet dynamics because the peculiar annular imbibition surrounding the droplets placed on the nanoporous thin film surfaces. We added in the revised version a sketched introductory Fig. 1 and a summarizing paragraph of the system in the introduction section (please see page 3 in main text) clarifying in more detail the key role of the nanoporous surface in order to achieve an intelligent droplet interplay via underlying chemical messages. In addition, we show control experiments on non-porous surfaces where, as expected, the droplet dynamic behavior does not occur (please see page 7 in main text, together with Supplementary Fig. 3 and Supplementary Fig. 4). Following the reviewer's recommendation, we also included further explanations about the formation mechanism of the wetted annulus in the nanopore network around the droplets (please see pages 5 and 6 in main text).

(2) Would it make sense to vary not only the liquid constituents of the droplets but also the pore morphology and pore size to achieve distinct droplet merging dynamics in the future?

Answer-Action taken: We are grateful for this comment that encouraged us to give a perspective of considering the porous system in relation to the behavior of the drops. We performed additional experiments that preliminarily explore the role of nanopore morphology and even surface modifications on droplet interplay and briefly discussed the results, which are now added to the revised version. With respect to varying the pore network structure, we now show that the complex emerging dynamic between the droplets is not limited to one specific nanopore configuration (please see page 21 in main text and Supplementary Video 8). We as well show that nanopore surface modifications can alter both fluidic and chemical inter-droplet communication and consequently affect the dynamical response of the droplets (please see pages 20 and 21 in main text, together with Supplementary Fig. 7 and Supplementary Video 7). We then also remark the potential of nanopore system tuning (i.e., nanopore size and network morphology, framework composition, and pore surface functionalization) to achieve a custom design of droplet dynamic interactions in the future.

(3) So far only the thermal diffusion time scales are estimated. It would be great to get also a quantitative or semi-quantitative understanding of the spreading and merging kinetics of the droplets. I would guess that it sensitively depends also on the fluid imbibition and fluid transport dynamics in the nanoporous films. Is it possible to relate the observed kinetics with simple models for transport of mesoporous media, e.g. based on the Darcy description?

Answer-Action taken: This is a challenging observation that stimulated us to further investigate the possible mechanisms for the protrusion formation and spreading. We carefully considered the physicochemical phenomena taking place at the moving contact line and, after modelling and calculations, concluded that the advance of the liquid front is driven by the Marangoni effect that originates at the H₂O₂-drop contact line, where the liquid surface tension is reduced by the chemical reaction. The Marangoni stress acts at the edge of the macroscopic meniscus and drag the liquid from the H₂O₂ droplet towards the KI-drop where γ is higher, thus generating the protrusion. In this framework, we derived a simple but physically sound model that quantitatively describes the kinematics of the observed process. We thank the Reviewer for this comment that certainly helped us to improve the theoretical description of our work.

Finally, to specifically answer the last Reviewer's question, one should note that fluid transport through the porous film (Darcy hydrodynamic regime) is taken into account to describe the formation of the wet annulus around the drops. The spreading phenomenon would take place preponderantly over the infiltrated nanoporous surface.

In the revised manuscript, we have included the novel calculations together with model foundations, description, and references about the protrusion kinematic (please see pages 12 to 15 and 19 in main text). In particular, please see the new Supplementary Fig. 5a, Supplementary Fig. 5b, and Supplementary Fig. 5c, and the related text. The description of the wet annulus formation has also been largely improved in the revised manuscript; please see the upgraded Fig. 2 and the related text below (page 6 in main text).

Reviewer #2 (Remarks to the Author):

Pizarro et al report in their manuscript “Droplets in Underlying Chemical Communication Recreate Cell Interaction Behaviors” on droplet interaction mediated by an underlying mesoporous surface. This is a very timely topic of high interest to a broad readership. I would support publication in Nature Communication, however not in its present form. Normally, I don’t suggest as peer reviewer additional experiments to be done. However, given the potential of this work and the high standard of Nature Communications, I would ask the authors to consider the below comments in a revised manuscript:

Answer: We would like to thank the Reviewer for his/her careful reading and positive comments on our paper

- The introduction is in my view too brief and does not set the scene for the state of the art and the novelty provided by this work. While the milestone papers are mentioned, more detail should be offered including a broader portfolio of relevant papers and the current knowledge gap.

Answer-Action taken: As suggested by the reviewer, we expand the introduction by making explicit mention of important topics related with the work (active matter and out-of-equilibrium scenarios, among others), and incorporated a paragraph summarizing the relevant aspects and the novelty of the work (please see pages 2 and 3 in main text). We included as well several new references in order to better position the work in the field (23 new references in relation to the introduction section). We also added along the revised version highly relevant papers on which we based new discussions (21 other references).

- In my view, the key novelty of this paper is the fact that the underlying mesoporous layer is able to mediate droplet interaction through mesopore-driven fluid transport beyond the topological footprint of the macroscopic droplet. In my view the result shown in Figure S1 is key and should therefore be included in the main manuscript.

Answer-Action taken: We agree with the Reviewer that a main novelty of the work is the underlying nanopore-driven mediation between droplets. Therefore, we now included in the revised manuscript the former Figure S1 as the new Fig. 2.

- While the evidence provided in Figure S1 is compelling, the extent of pore filling could be spatially mapped by either microspectroscopy (offering spectroscopic sampling of approx. 10 μ m sized spots under an optical microscope) or by spectroscopic ellipsometry with microspot optics (spatial resolution approx 60 μ m).

Answer-Action taken: Firstly, we are grateful for the positive opinion on the figure. Although the reviewer's request is interesting for further characterization of nanopore infiltration, we do not currently dispose of these techniques. On the other hand, here we are interested in presenting a fresh scenario in droplet management and thus the proposed experiments on the nanopore infiltration are beyond the scope of this particular communication. Notwithstanding, we now performed in the new Supplementary Fig. 1 of the revised version an analogous spatial mapping of the extent of pore filling by using light intensity profile analysis under an optical microscope (please also see the text below the Supplementary Fig. 1 for method description).

- In my view, it would be quite valuable to investigate the role of surface modification of the mesopores (e.g. through grafting of octadecyltrichlorosilane), which will affect the wetting of the pores and therefore also the droplet communication. A simple control experiment would be to study droplet communication on a solid / non-porous support. Establishing the role of pore functionalization and porosity in general would also provide means for programmable surfaces.

Answer-Action taken: We thank the reviewer for this interesting suggestion. Firstly, we show now control experiments on solid / non-porous supports. When the droplets were placed on the silicon substrate (in the absence of the nanoporous film coating) or on a dense titania thin film (in the absence of template), no droplet responses were observed (please see page 7 in main text, together with Supplementary Fig. 3 and Supplementary Fig. 4). While here we are interested in presenting for the first time how an underlying chemical communication permits to generate complex macroscopic droplet dynamics, we

appreciate the reviewer's suggestion that stimulated us to undertake a preliminary exploration of pore functionalization and porosity in the platform. We now added exploratory experiments on films with surface modified nanopores or with another nanopore morphology. We performed two different surface modifications: grafting of either octyltriethoxysilane (OTES) to introduce hydrophobic alkyl chains on surface or (3-aminopropyl)triethoxysilane (APTES), a frequently used amino functionalization. We show that nanopore surface modifications can alter both fluidic and chemical communication between the droplets and consequently affect the dynamical behavior of the droplets (please see pages 20 and 21 in main text, together with Supplementary Fig.7 and Supplementary Video 7). Concerning porosity, we show that the complex inter-droplet responses are not limited to one specific nanopore configuration (please see page 21 in main text and Supplementary Video 8). We include the new results in the revised version and point out the potential of surface functionalization as an alternative to achieve a custom control of droplet dynamics due to the wide palette available of organic or metallo-organic functions.

- In my view, the authors have found a highly interesting phenomena that is worth reporting on a high profile platform. However, some of the analogies appear a little far reached. It would be quite useful to understand why the authors consider their work to be biomimetic. While I don't challenge the role of ions and reactive oxygen species in cell communication, the novelty of the work reported herein is about substrate mediation. It would be interesting if the authors could provide a better link to cellular behaviors.

Answer-Action taken: We thank again the Reviewer for the positive opinion on our work. We agree with the Reviewer that a main novelty of this paper is the nanopore layer mediation between droplets and therefore emphasized this concept along the revised manuscript. In addition, taking into account the reviewer's comment, in this new version we also provide a better discussion of the mechanical-functional analogies between our minimalist platform with living systems. Macroscale dynamic phenomena emerge in cell systems when locally consume chemical energy in the microscale to produce moving forces. For example, many biological functions, such as cytoplasmic streaming, morphogenesis, and cell migration, are driven by active stresses that emerge from active components localized within the cell. In particular, the cell protrusions are owing to active stress driven actin filaments undergoing polymerization that focalizedly curve the cell

membrane forward. While our minimal platform is not intended to represent such evolved biomolecular mechanism, it is interesting to note that the dynamic droplet responses bear resemblance to protrusion emission and also phagocytic actions in biological cells. Interestingly, it is even worth noting that the origin of shape-morphing droplet can be traced back to a rudimentary way of localized active stresses (i.e., a local chemical activity that induces an active stresses - chemically induced local Marangoni stress -). Our results indeed reveal that complex droplet dynamic behaviors reminiscent of cell morphogenesis can arise despite the relatively simple active matter framework employed. It should be also noted here that our platform further combines a set of physico-chemical ingredients inherent to cell functional features: 1) specific stimulus-response interactions, 2) role differentiation and 3) transformation of chemical energy generated in a localized space into mechanical work. These discussions are now included in the revised manuscript to better clarify the link between our simple droplet framework and cellular behaviors (please see page 22 in main text).

Reviewer #3 (Remarks to the Author):

The work presented by the authors is an original attempt towards mimicking a basic cellular process, such as phagocytosis, via a competition between chemical reactions of two nanoconfined fluids, and the mechanical pinning energy associated with the drop stability of one of the interacting fluids when placed on a nanoporous membrane. The novelty of this study lies in simulating an attacker–victim like situation analogous to the entrapment of a protein/pathogen by a cell/neutrophil guided solely by an autonomous chemospecific interaction, without the requirement of any external stimuli or pre-existing trajectory. The authors determined conditions, in terms of concentration and the chemical nature of the constituents, yielding favorable conditions for the intended chemo-specific response. The authors have also provided a reasonable thermodynamic basis to rationalize the attacker–victim like interaction, within which the aq. H₂O₂ droplet utilizes a trajectory provided to it by the advancing annulus of the aq. KI drop.

The novelty in the scope and objectives of this study potentially makes it a fresh contribution to the existing literature in this domain, which is well cited in the manuscript. The experimental methodology used and the analysis and interpretation has a logical progression and appears clear enough for reproduction and customization. However, the investigations on the system should significantly be expanded, in terms of both experimental results and analysis. In its present state the paper lacks a clear presentation of the physical principles giving rise to the observed behavior, and would greatly be improved with e.g. a basic quantitative model explaining how to manipulate the interaction behaviors as a function of well-identified external dimensionless parameters. Most likely, a significant revision would be necessary to warrant publication in Nature Communications.

Answer-Action taken: We appreciate the positive opinion of the Reviewer on the experimental work, and admit that the physical principles of the wetting phenomenon were missing in the manuscript. After this comment, we have effectively considered a plausible mechanism for the protrusion formation and spreading. We considered the physicochemical phenomena taking place at the advancing contact line and, after modelling and calculations, concluded that the motion of the liquid front is driven by the Marangoni effect that originates at the H₂O₂-drop contact line, where the liquid surface tension is locally reduced by the chemical reaction. The Marangoni stress acts at the edge of the macroscopic meniscus and pulls the H₂O₂-droplet contact line towards the KI-drop where γ is higher, thus deploying the protrusion. From this analysis, we derived a simple though physically founded model, which quantitatively describes the kinematics of the wetting process in terms of well-identified parameters, such as the surface tension and the fluid viscosity. We think that the theoretical description of our work has been notably improved after this Reviewer's comment.

In the revised version, we have included these calculations together with model foundations, description, and references. In particular, please see Supplementary Fig. 5a, Supplementary Fig. 5b, and Supplementary Fig. 5c and related text in pages 12 to 15 and 19 of revised manuscript.

In addition to the general comments above, please find below some remarks and questions that would need to be addressed.

1. The paper would be clearer with an introductory figure showing a sketch of the system of interest with drawings of the basic processes at play (e.g. based on Figure 2) and summarizing the study. Currently, most of the figures are microscope images and/or graphs and feel rather technical.

Answer-Action taken: We appreciate the reviewer's comment that helped us clarify and illustrate our descriptions. We now include an introductory sketch in the new Fig. 1 and a summary of the basic processes at play in the introduction section (please see page 3 in main text).

2. The structural aspects of the porous host matrix have not been sufficiently elaborated. The existence of a “neck” and “pore” gives two indications; the “necks” being interconnections between the “pores” or the existence of ink-bottle geometries, wherein the pores get narrower (“necking”) towards their openings. Could the authors provide further structural information on their porous membrane characterization?

Answer-Action taken: We agree that the information on structural characterization was somewhat insufficient in the previous version of Supplementary Information. The environmental ellipsometric porosimetry (EEP) is a powerful technique that allows assessing the porosity of the nanoporous films that is difficult to determine otherwise. The type-IV isotherm with a H1-like hysteresis loop (according to IUPAC classification) observed in the water adsorption-desorption measurement is indicative of the presence of open channel type pores. The delayed closure of the hysteresis loop in turn indicates pore constrictions (necks) along the main structural mesoporosity. Pore size distribution were estimated from the adsorption branch and desorption was assumed to be representative of the neck sizes. Structural information on the porous characterization is now expanded and clarified in the revised version (please see new Supplementary Fig. 2 and related text below).

3. The transport of liquids can be strongly modified under nanoporous confinement and the geometry of confinement also plays a significant role, for example, with introduction of anisotropy in the diffusion coefficient. Can the authors comment on whether these

effects are relevant in their study (e.g. to explain the timescales involved in the experiments)?

Answer-Action taken: In fact, the structure of the nanoporous material affects the liquid transport through the film, which affects the dynamics of the wet annulus. In turn, the existence of the wet annulus around the first drop is critical to set the scenario for the chemical reaction occurring in the second drop border. Nevertheless, the spreading phenomenon itself takes place preponderantly over the already infiltrated nanoporous surface.

The formation of the wet annulus is now better described; see the upgraded Fig. 2 and the related text below. Please also note that the description of spreading phenomenon leading protrusion evolution has been much improved in the revised manuscript, as mentioned above.

4. What is the role played by the KI concentration? Will the phagocytic entrapment of aq. KI by aq. H₂O₂ be still observed if the aq. KI concentration is raised to 2% w/w, or reduced to 0.5% w/w for example?

Answer-Action taken: We thank the reviewer for this valuable comment that led us to evaluate another system control variable. Although a similar phagocytic engulf was observed when increasing the concentration to 2 wt%, an interesting scenario was observed when decreasing the KI concentration. By decreasing the KI concentration to 0.5 wt%, only a short roam of the protrusion occurred on the outline of the KI drop before coupling and no attraction between both droplets was observed in the case of 0.1 wt% KI concentration (please see Supplementary Videos 5 and 6). These results are now discussed in page 19 of the revised version.

5. Could the barrier resistance resulting from KCl be simply a consequence of its low concentration? Could a possible oxidation of the chloride ion to chlorine gas by H₂O₂ have led to an attacker–victim situation as well, given the concentration of KCl was an order of magnitude higher?

Answer-Action taken: We perform the control experiment on the behavior of the H₂O₂ droplet by replacing the KI droplet with an equivalent KCl solution (1 wt%) to assess the

chemo-specificity of the attacking droplet response based on iodine-catalyzed H_2O_2 decomposition. In fact, a "barrier" effect was observed between both droplets in the absence of iodine to catalyze H_2O_2 decomposition. In accordance with the reviewer's comment that another chemical action (oxidation of the chloride ion to chlorine gas by H_2O_2) could be possible in the case of higher (order of magnitude) KCl concentration and also lead to an attacking response, we perform the equivalent experiment with a KCl droplet concentration of 10 wt%. For this KCl concentration, a similar "barrier" effect was observed, as shown in the Figure below:

6. As mentioned in the manuscript, and visible from the figures and supplementary videos, the protrusion width clearly has an impact on the connection time. However, the factors leading to differences in the protrusion widths is not satisfactorily explained, and hence require further elaboration.

Answer-Action taken: We thank the reviewer for this suggestion. The different overlapping between the H_2O_2 droplets with the annular signaling from the KI droplets is the factor that produce the differences in the protrusion widths. We have now added an explanatory paragraph in pages 15 to 16 of revised version and a schematic Fig. 5a to clarify this subject.

7. The speed of protrusion as shown in Figure 1b increases very sharply up to 13 s and then starts decreasing gradually, while no sharp changes are observed in the position. How do the authors justify it?

Answer-Action taken: This is an interesting comment since it has to do with basic physics. Firstly, let us note that we have arbitrarily chosen the origin of the coordinate system ($x=0$, $t=0$). At this time the drop contact line is steady and then the Marangoni flow regime has a sort of induction time, which correspond to the formation stage of the

concentration and temperature gradients, after the onset of the chemical reaction (please see the proposed mechanism in the revised version). Thus an initial increase in velocity (from still to being in motion) in this transitional stage is physically expected. The posterior velocity decrease is because the surface tension gradient ($\Delta\gamma/x_f$), which is the driving force of the contact line displacement, decreases with the advance of the fluid front position, as quantitatively described by the proposed model. These explanations have been included in the revised manuscript (please see page 14 to 15).

Please find below some additional, minor recommendations.

1. The SEM image of the porous membrane is of the 2D plane perpendicular to the pore-axis. A cross-sectional SEM/TEM image of the plane parallel to the pore axis will also be quite informative.

Answer-Action taken: A cross-sectional SEM view of the nanoporous film is now provided in Supplementary Fig. 2.

2. The caption for Fig 1 (a) has a lot of text that is better suited for the main text, Panels should be numbered (e.g. i, ii, iii, etc), and more precisely referred in the main text. This comment holds for all figures with multiple panels in a single figure sub-part.

Answer-Action taken: Thanks for the suggestion. Part of the previous legend of Figure 1 has now been incorporated into the main text. The subpanels of all multi-panel Figures have now been numbered and hence more precisely referenced in the main text.

3. The quantity E_{ch} needs to be defined.

Answer-Action taken: Thanks for the remark. This is now defined in the revised version (please see page 10 in main text).

4. In page 8 of the manuscript, it is mentioned that V_d/A_d is approx. equal to 10^{-4} , and the reader's attention is drawn towards Figure 1 for reference. However, looking at

Figure 1 this aspect does not seem apparent. Please consider elaborating Figure 1 suitably to address this issue.

Answer-Action taken: We thank the reviewer for pointing out this ambiguous sentence. Being respectively V_d and A_d the volume and the contact area of the drop, we simply estimate the V_d/A_d ratio considering: $V_d = 2 \mu\text{l}$ (typical drop volume employed) and a A_d estimated by drop contact area extrapolation from current Fig. 3 (former Figure 1). We have now clarified the sentence in the revised version (please see page 11 in main text).

5. “Likewise, the connection time between the droplets (i.e., the time from protrusion nucleation to droplet coupling) is linked to protrusion size: for example, increasing two-fold the width of the protrusion (see Figure S3 in SI) reduces droplet connection time from 125 to 40 s (See Figure 1 and Figure 3 for the 30 % H₂O₂ sample).” (Page 10 of the manuscript). A direct reference to Figure S3 does not give any clear information about the connection time between the droplets. I suggest elaborating Figure S3 more, such that it can be directly correlated with Figures 1 and 3 in the main manuscript.

Answer-Action taken: We thank the reviewer for the suggestion. A more elaborate Figure S3 has now been included as Fig. 5b in the revised version, directly illustrating the reduction on the connection time related to the width of the protrusion.

REVIEWER COMMENTS

Reviewer #1 (Remarks to the Author):

The authors addressed all of my criticisms and suggestions of the first reviewing round. I recommend a publication of this beautiful piece of work in Nature Communications without hesitation.

Reviewer #2 (Remarks to the Author):

I have now carefully reviewed the revised manuscript and can confirm that all my comments have been adequately addressed. With the revisions, this manuscript has now further improved in my view and displays some highly interesting science that in my view deserves publication in Nature Communications.

Maybe one marginal comment:

The authors are referring to their results at the end of the introduction: This concept of an underlying nanoporous layer to mediate an intelligent droplet interplay (chemospecific inter-droplet role-difference autonomous operation - sketched in Fig. 1) indeed successfully works as described in the present paper.

In my view I would leave this as a more open statement (without anticipating the results) to transition into the following paragraph that puts their work in context. But may be down to personal taste.

Congratulations.

Reviewer #3 (Remarks to the Author):

The authors have significantly elaborated the mechanisms associated with the droplet engulfment phenomena discussed in this manuscript. The use of simple macroscopic models combining concepts of Marangoni effect and the interplay of liquid imbibition and evaporation phenomena to explain the protrusion growth and annulus development, and their correlation with the underlying chemical processes have helped enrich the discussion.

The authors have responded to all the questions thoroughly and deserve appreciation for reporting additional experiments (comparison with bare substrates, additional surface treatments, other pore morphologies etc.) and expanding on ellipsometric porosimetry studies. This additional data has improved the understanding of the phenomena at play and the characterization of the host porous matrix's architecture.

One of the questions concerning the effect of nanoscale confinement on the transport properties of the guest molecules has not been satisfactorily answered (or maybe not well understood), though its effect on the general objectives of this manuscript is not critical. It is well established that the self-diffusion coefficient of water can decrease by up to an order of magnitude in silica nanopores (approximately or less than 10 nm diameter) and this can be particularly aggravated by the "necks" between individual pores. This reduced spatial geometry of diffusion and the increasing role of the interfacial interaction between the guest molecules (water and hydrogen peroxide here) and host matrix can potentially increase the values of $\langle \Delta z \rangle / 2D$ to 30 s (instead of 3 s), indicating that the time scale of protrusion evolution could be extremely slow in the absence of chemo-specific activity. Looking at the slow kinetics of the supplementary videos, it appears that the above mentioned confinement induced effects might indeed be present.

Some additional minor remarks :

- Some of the text in Fig. 6b is mirrored, which makes reading difficult.

- In reference S8 in the Supplemental Material, the authors are missing.

Overall, given the extent of systematic improvements made by the authors, the manuscript seems suitable for publication in Nature Communications.

====Additional Reviewer #3 comment about the nanoscale confinement:

In their discussion in page 12 of the revised manuscript, the authors have described the physical conditions that allow a gradual progress of the protrusion process with minimal chances of local overheating by comparing the time scale of molecular diffusion of the guest molecules with the time scale of thermal diffusivity. Since the latter is two orders of magnitude lower, the authors claim that the probability of local overheating interfering in the progress of the physico-chemical processes of protrusion is going to be limited, which is a fair argument. Our comment was on the estimation of the diffusion time scales, which in our understanding should be an order of magnitude higher, given the well established knowledge that spatial confinement of nanometric dimensions can slow down the molecular diffusion by an order of magnitude for most liquids, in particular water. This happens due to the increased competition between the guest-guest and guest-host interactions as the confinement dimensions reach quite close to molecular dimensions (especially at the pore necks) and the bulk continuum properties tend to break down. From our observation, the progress of the protrusion is quite slow and the dynamics might better correspond to slow down in molecular diffusion (characteristic time scale being 30 s instead of 3s) resulting from. However, this goes well with the authors' claim as it strengthens the point further that the effect of local overheating will be even lesser than expected. The authors can use a diffusion coefficient of $10^{-10} \text{ m}^2/\text{s}$ for a rough estimation instead of $10^{-9} \text{ m}^2/\text{s}$. As a suitable reference they can cite Phys. Chem. Chem. Phys., 2020, 22, 13989--13998 and J. Phys. Chem. C 2013, 117, 3330–3342.

Detailed answer to the reviewers' comments:

In the following, the reviewers' comments are typeset in *italics* while our replies appear in **blue regular font**.

Reviewer #1 (Remarks to the Author):

The authors addressed all of my criticisms and suggestions of the first reviewing round. I recommend a publication of this beautiful piece of work in Nature Communications without hesitation.

Answer: We appreciate the reviewer's comments on our revised manuscript, as well as the recommendation of this manuscript to be published in Nature Communication.

Reviewer #2 (Remarks to the Author):

I have now carefully reviewed the revised manuscript and can confirm that all my comments have been adequately addressed. With the revisions, this manuscript has now further improved in my view and displays some highly interesting science that in my view deserves publication in Nature Communications.

Maybe one marginal comment:

The authors are referring to their results at the end of the introduction: This concept of an underlying nanoporous layer to mediate an intelligent droplet interplay (chemospecific inter-droplet role-difference autonomous operation - sketched in Fig. 1) indeed successfully works as described in the present paper.

In my view I would leave this as a more open statement (without anticipating the results) to transition into the following paragraph that puts their work in context. But may be down to personal taste.

Congratulations.

Answer: We thank the Reviewer for the positive comments on our revised manuscript and also for considering that it deserves publication in Nature Communications.

Concerning to the marginal comment, we would like to preserve the sentence to highlight the goals achieved in the work.

Reviewer #3 (Remarks to the Author):

The authors have significantly elaborated the mechanisms associated with the droplet engulfment phenomena discussed in this manuscript. The use of simple macroscopic models combining concepts of Marangoni effect and the interplay of liquid imbibition and evaporation phenomena to explain the protrusion growth and annulus development, and their correlation with the underlying chemical processes have helped enrich the discussion.

The authors have responded to all the questions thoroughly and deserve appreciation for reporting additional experiments (comparison with bare substrates, additional surface treatments, other pore morphologies etc.) and expanding on ellipsometric porosimetry studies. This additional data has improved the understanding of the phenomena at play and the characterization of the host porous matrix's architecture.

Answer: We appreciate the positive reviewer's comments on our revised manuscript.

One of the questions concerning the effect of nanoscale confinement on the transport properties of the guest molecules has not been satisfactorily answered (or maybe not well understood), though its effect on the general objectives of this manuscript is not critical. It is well established that the self-diffusion coefficient of water can decrease by up to an order of magnitude in silica nanopores (approximately or less than 10 nm diameter) and this can be particularly aggravated by the “necks” between individual pores. This reduced spatial geometry of diffusion and the increasing role of the interfacial interaction between the guest molecules (water and hydrogen peroxide here) and host matrix can potentially increase the values of $\langle \Delta z \rangle^2 / 2D$ to 30 s (instead of 3 s), indicating that the time scale of protrusion evolution could be extremely slow in the

absence of chemo-specific activity. Looking at the slow kinetics of the supplementary videos, it appears that the above mentioned confinement induced effects might indeed be present.

Additional Reviewer #3 comment about the nanoscale confinement:

*In their discussion in page 12 of the revised manuscript, the authors have described the physical conditions that allow a gradual progress of the protrusion process with minimal chances of local overheating by comparing the time scale of molecular diffusion of the guest molecules with the time scale of thermal diffusivity. Since the latter is two orders of magnitude lower, the authors claim that the probability of local overheating interfering in the progress of the physico-chemical processes of protrusion is going to be limited, which is a fair argument. Our comment was on the estimation of the diffusion time scales, which in our understanding should be an order of magnitude higher, given the well established knowledge that spatial confinement of nanometric dimensions can slow down the molecular diffusion by an order of magnitude for most liquids, in particular water. This happens due to the increased competition between the guest-guest and guest-host interactions as the confinement dimensions reach quite close to molecular dimensions (especially at the pore necks) and the bulk continuum properties tend to break down. From our observation, the progress of the protrusion is quite slow and the dynamics might better correspond to slow down in molecular diffusion (characteristic time scale being 30 s instead of 3s) resulting from. However, this goes well with the authors' claim as it strengthens the point further that the effect of local overheating will be even lesser than expected. The authors can use a diffusion coefficient of 10^{-10} m²/s for a rough estimation instead of 10^{-9} m²/s. As a suitable reference they can cite *Phys. Chem. Chem. Phys.*, 2020, 22, 13989--13998 and *J. Phys. Chem. C* 2013, 117, 3330–3342.*

Answer-Action taken: We thank the reviewer for the thorough explanations. While we agree with the reviewer that this detail is not critical to the objectives of this manuscript, we added in this revised version a clarification on this issue.

We are aware that in several nanopore/neck systems, the diffusion coefficient values are lower by up to an order of magnitude than in bulk solution because of confinement effects, and that these differences are due to an increased competition between the molecule-molecule with the molecule-surface interactions, as the references cited by the reviewer say. Taking into account this limiting case, the estimated diffusion time in our system can be indeed increase to 30 s. Therefore, if the diffusion coefficient of any species were one order of magnitude lower, this will mean that a local overheating effect is even less expected, further strengthening the point

of our claim as sharply stated by the reviewer. As also the reviewer points out, the slow protrusion progress dynamics might better correspond to slow down in molecular diffusion due to the confinement-induced effects.

We now included in page 12 of the revised version a short text in order to clarify the issue. We agree that the suggested references are relevant, so also we added them.

Some additional minor remarks:

- Some of the text in Fig. 6b is mirrored, which makes reading difficult.

Answer-Action taken: Thanks for the remark. We have now modified the Figure 6b following the reviewer suggestion.

- In reference S8 in the Supplemental Material, the authors are missing.

Answer-Action taken: Thanks for the remark. We have now added the authors in the reference S8 in the Supplementary Information.

Overall, given the extent of systematic improvements made by the authors, the manuscript seems suitable for publication in Nature Communications.

Answer: We thank the reviewer for considering that our manuscript is suitable for publication in Nature Communications.